# Transport of Thiophanate Methyl in Porous Media in the Presence of Titanium Dioxide Nanoparticles

Anthi S. Stefanarou [1], Vasileios E. Katzourakis [1,2], Fenglian Fu [3], Anastasios A. Malandrakis [1] and Constantinos V. Chrysikopoulos [1,2,*]

1  School of Chemical and Environmental Engineering, Technical University of Crete, 73100 Chania, Greece
2  Department of Civil Infrastructure and Environmental Engineering, Khalifa University of Science and Technology, Abu Dhabi 127788, United Arab Emirates
3  School of Environmental Science and Engineering, Guangdong University of Technology, Guangzhou 510006, China
*  Correspondence: constantinos.chrysikopoulos@ku.ac.ae

**Abstract:** Human activities in modern life are contributing significantly to global environmental pollution. With the need for clean drinking water ever increasing, so does the need to find new water-cleaning technologies. The ability of nanoparticles (NPs) to remove persistent pollutants from aqueous solutions makes them very important for use in water treatment technology. Titanium dioxide ($TiO_2$) is recognized as an NP with unique optical, thermal, electrical, and magnetic properties and is widely used as an adsorbent material. Due to the extensive use of pesticides, their removal from the aquatic environment has gained widespread attention from the scientific community. In the present work, the transport of pesticide thiophanate methyl (TM), as well as the cotransport of TM and $TiO_2$ nanoparticles, in a water-saturated column packed with quartz sand under various water conditions were investigated. Several ionic strengths (1, 10, 50, and 100 mM) and pH values (3, 5, 7, and 10) were examined. The results from the transport experiments were fitted and analyzed with the use of the ColloidFit software, while the results from the cotransport experiments were fitted with a modified version of a recently developed mathematical cotransport model. The results of this study suggested that the lowest mass recovery rate was for the cotransport experiments with the addition of NaCl. Furthermore, it was shown that TM has a weak affinity for sand but a relatively strong affinity for $TiO_2$ at high ionic strength and acidic pH, probably accounting for the reduced mass recovery of TM in cotransport experiments.

**Keywords:** nanoparticles; titanium dioxide; pesticides; thiophanate methyl; cotransport; porous media; quartz sand; column experiments

## 1. Introduction

The world population is growing and is expected to reach nearly 10 billion by the year 2050 [1]. Consequently, the existing agricultural system is pressured to increase food production in order to meet the demand. Therefore, the use of pesticides worldwide has increased in order to control or exterminate the development of plant pests and diseases, including mites, insects, and nematodes [2–4]. In countries where pesticides are widely used, such as Brazil, the world's largest consumer of these substances, the average pesticide use is greater than 10 L per hectare, which corresponds to a mean exposure of 4.5 L of pesticides per capita per year [5]. As a significant portion of pesticides used cannot be absorbed by crops, they are ultimately released into the aquatic ecosystem, which in turn leads to adverse effects on humans and aquatic organisms [6,7]. Human exposure to pesticides can occur directly through occupational exposure, as in the case of farmers, or indirectly through environmental exposure to air, water, and soil, as well as through the consumption of food containing pesticide residues [8,9].

Nanoparticles (NPs) are materials that have one dimension in the 1–100 nm range [10]. Therefore, at least one dimension of a nanoparticle is about one hundred thousand times smaller than the diameter of a human hair or the thickness of a paper sheet [11]. As a result of their unique physicochemical properties, NPs are frequently employed in numerous applications, which include medical, catalysis, electronics, cosmetics and personal products, food supplements, coatings, pharmaceuticals, clothing, sportswear, and environmental applications [12–15], resulting in a massive increase of their use [16,17]. Titanium dioxide ($TiO_2$) is insoluble in water, very stable, with a high refractive index. As a result of its excellent properties to filter UV radiation, inhibit corrosion, and exhibit excellent antimicrobial activity, $TiO_2$ has found uses in food coloring, paints, inks, coatings, and personal products (toothpaste, face powders) and is commonly found in rutile, anatase, and brookite crystalline forms [18,19]. Due to the incorporation of $TiO_2$ NPs in consumer products, their potential release into the environment is increasing exponentially and may cause adverse health effects to humans and aquatic organisms.

Thiophanate Methyl (TM) is a benzimidazole fungicide, which has been used against a wide range of plant diseases in various crops, both pre- and post-harvest, since 1973. TM is also used in pastures and ornamental plants, greenhouses, and nurseries [20]. TM is classified as grade IV or low toxicity [21]. However, TM and its metabolite, methyl benzimidazole-2-yl carbamate (MBC), are known to cause histopathological damage to the thyroid and adrenal glands of lizards [22] and to the kidneys, liver, and blood of rats [23].

The migration of pesticides in environmental systems can easily occur through surface runoff, rinsing, drainage, and spraying [24]. Consequently, the removal of pesticides from the environment is a very important task that has received widespread attention. The removal of pesticides from aqueous solutions can be achieved by numerous chemical and physical treatments, such as adsorption, advanced oxidation, membrane filtration, phytotherapy, bioremediation, and activated sludge [25]. However, most of the available pesticide removal techniques are of high cost, limited flexibility, and low efficiency [26], making the removal of pesticides from aqueous solutions with adsorption onto low-cost materials a promising technique.

Adsorption is a process that is used to effectively remove hazardous organic as well as inorganic impurities from aqueous solutions [27]. Adsorption can be a simple and inexpensive wastewater treatment process if adsorbents that are abundant in nature are employed [28]. Since the adsorption process is associated with surface reactions, the physical and chemical properties of the adsorbents used can significantly affect the adsorption efficiency [26]. Furthermore, nanoparticles and carbon-based adsorbents are known to exhibit high efficiency in removing pesticides from aqueous systems [29]. To the best of our knowledge, although various procedures for pesticide removal using nanoparticles have been proposed in the past [30,31], the use of $TiO_2$ nanoparticles for TM removal under different pH and ionic strength conditions has not been previously examined. The present study focuses mainly on the effect of $TiO_2$ nanoparticles on the removal of the TM pesticide in water-saturated porous media and explores the behavior of $TiO_2$ and pesticide cotransport. Furthermore, the experimental data collected in this study were successfully fitted with a cotransport numerical model previously published in the literature, which was properly modified to account for the irreversible sorption of TM onto $TiO_2$ nanoparticles. Finally, columns packed with quartz sand were used as a typical filtration system.

## 2. Materials and Methods

The nanoparticles used in the experiments performed in this study were titanium dioxide ($TiO_2$) anatase (Aldrich 637254-50G, purity N 99.9%, size < 25 nm, St. Louis, MO, USA). The required $TiO_2$ NPs suspensions were prepared by mixing 0.1 g of $TiO_2$ powder in 1000 mL of Milli-Q distilled deionized water (dd$H_2O$). The suspensions were then placed in an ultrasonic bath (sonication bath Elmasonic S 30/(H), Elma Schmidbauer GmbH, Singen, Germany) for 30 min to achieve a uniform dispersion of $TiO_2$ NPs.

The pesticide used in the experiments was thiophanate methyl (TM, chemical structure shown in Figure 1 [32]) of 70% purity (Sigma-Aldrich 45688). The required TM stock solution (10 mg/L) was prepared by transferring 14.3 mg of TM in a 1000 mL volumetric flask and adding ddH$_2$O to the volumetric flask mark. Subsequently, the TM stock solution was sonicated for 10 min.

C$_6$H$_4$(NHC(S)NH(CO)OCH$_3$)$_2$

**Figure 1.** Chemical structure of thiophanate methyl [32].

Batch adsorption experiments were performed to investigate the interaction between the TM pesticide with quartz sand (Filcom, Sibelco Co., Wessem, The Netherlands). The sand had a specific density of 2.6 gr/cm$^3$ and a size range from 400 to 800 μm. The quartz sand was cleaned with 0.1 M HNO$_3$ and 0.1 M NaOH by following previously established procedures [33,34]. All batch experiments (static and dynamic) were conducted in 20 mL Pyrex glass screw-cap tubes (Fisher Scientific, Waltham, MA, USA) under controlled conditions at four different pH values (pH = 3, 5, 7, and 10) and at four different ionic strengths (Is = 1, 10, 50, and 100 mM) with initial TM concentration C$_0$ = 10 mg/L, at room temperature. For each batch experiment, 20 screw-cap tubes were used (10 for the static and 10 for the dynamic group). Each tube contained 14 mL of TM solution with 14 g of quartz sand. For the static experiments, the glass screw-cap tubes were stowed in a tube holder, whereas for the dynamic experiments, the glass screw-cap tubes were placed in a typical rotator (Selecta, Agitador orbit), which was revolving at 12 rpm in order to maintain a thorough mixing of the quartz sand and the TM suspension. At preselected times (5, 15, 30, 45, 60, 90, 120, 150, 180, and 240 min), one screw-cap tube was taken at random from each of the two groups (static and dynamic). Subsequently, the collected samples were centrifuged at 30,000 rpm for 10 min in a microcentrifuge to settle any possible suspended particles.

The laboratory-scale flow-through experiments were conducted in a glass column with an inner diameter of 2.5 cm and a length of 30 cm. The column was filled with quartz sand and was placed horizontally in order to avoid possible gravity effects [35]. For each flow-through experiment, freshly cleaned quartz sand was used. The column was packed by inserting small incremental volumes of the quartz sand under gentle vibration in order to ensure uniform packing [36]. The packed column was initially saturated with ddH$_2$O. The experimental procedures, as well as the sample collection methodology employed, are described in earlier works [37,38]. Two series of flow-through experiments were performed under controlled conditions for four different pH values (pH = 3, 5, 7, and 10) and at four different ionic strengths (Is = 1, 10, 50, and 100 mM) at room temperature. For the first set of experiments, the injected fluid contained only TM (10 mg/L), whereas in the second set of experiments, the injected fluid contained both TM (10 mg/L) and TiO$_2$ NPs (100 mg/L). The solution ionic strength was adjusted with the addition of NaCl, while the solution pH was adjusted to the desired value with the addition of either 0.1 M HCl or 0.6 M NaOH. The size and the dynamic zeta potential of the NPs were determined with a zeta sizer (Nano ZS90, Malvern Instruments, UK).

The concentrations of TM and TiO$_2$ NPs in the collected samples were determined by UV-Vis double-beam spectrophotometry (model UV-1900, Shimadzu) at a wavelength of 262 nm for TM and 287 nm for TiO$_2$ NPs. Three calibration curves were constructed. One calibration curve for the samples containing only TM, and two containing both TM and TiO$_2$ (one containing TM and traces of TiO$_2$, and the other containing TiO$_2$ and traces of TM),

which were used for the cotransport experiments where both TM and $TiO_2$ concentrations were determined.

The zeta potential ($\zeta$) and the hydrodynamic diameter ($d_H$) of the suspended $TiO_2$ NPs were measured with a zeta sizer (Nano ZS90, Malvern Instruments, Southborough, MA, USA). The zeta sizer employs Dynamic Light Scattering to measure the random movement of particles due to collisions by the molecules of the surrounding fluid (Brownian motion) and correlates this to obtain the size of the suspended particles. The zeta potential of $TiO_2$ of the initial solution measured was $\zeta = -35.6$ mV. The measured $\zeta$ and $d_H$ values for $TiO_2$ in the various experiments conducted here are listed in Table 1.

**Table 1.** Zeta potential and hydrodynamic diameter for $TiO_2$.

| Experimental Conditions * | | $TiO_2$ | |
|---|---|---|---|
| pH | Ionic Strength (mM) | $\zeta$ (mV) | $d_H$ (nm) |
| 3 | - | −25.1 | 315.6 |
| 5 | - | −25.7 | 317.0 |
| 7 | - | −28.0 | 326.6 |
| 10 | - | −42.3 | 567.3 |
| 7.5 | 1 | −36.0 | 389.3 |
| 7.5 | 10 | −41.0 | 278.0 |
| 5.4 | 50 | −29.9 | 386.3 |
| 5.1 | 100 | −27.0 | 1004.4 |

* Cotransport experiments.

## 3. Mathematical Modeling

### 3.1. Governing Partial Differential Equations

The mathematical model employed in this study is a modification of the cotransport model developed by Katzourakis and Chrysikopoulos [39]. The modified model accounts for the cotransport of non-aggregating nanoparticles and solute substances. The nanoparticles can either be found suspended in the aqueous phase, $C_n$ [$M_n/L^3$], or attached to the solid matrix, $C_{n*}$ [$M_n/M_{sm}$]. Solute concentrations are represented by $C_s$ [$M_s/L^3$]. Solutes may sorb onto suspended nanoparticles, $C_{ns}$ [$M_s/M_n$], or sorb onto the solid matrix, $C_{s*}$ [$M_s/M_{sm}$], or sorb on nanoparticles already attached to the solid matrix, $C_{n*s*}$ [$M_s/M_n$]. Note that the subscripts n, s, and ns represent nanoparticles, solutes, and nanoparticle-solute complexes, respectively. Additionally, $M_n$ represents the mass of nanoparticles, $M_s$ is the mass of solutes, and $M_{sm}$ is the mass of the solid matrix.

The transport of non-aggregating nanoparticles in one-dimensional, homogeneous, water-saturated porous media with developed one-directional uniform flow, accounting for kinetic attachment onto the solid matrix, is governed by the following partial differential equation [35,40]:

$$\frac{\partial C_n(t,x)}{\partial t} + \frac{\rho_b}{\theta}\frac{\partial C_{n*}(t,x)}{\partial t} - D_n\frac{\partial^2 C_n(t,x)}{\partial x^2} + U_n\frac{\partial C_n(t,x)}{\partial x} = F_n(t,x) \tag{1}$$

where x [L] is the Cartesian coordinate in the longitudinal direction; t [t] is time; $\theta$ [-] is the porosity of the porous medium; $\rho_b$ [$M_{sm}/L^3$] is the bulk density of the solid matrix; $D_n$ [$L^2/t$] is the hydrodynamic dispersion coefficient of the suspended nanoparticles; $U_n$ [L/t] is the interstitial velocity; and $F_n$ [$M_n/L^3t$] is a general form of the nanoparticle source configuration.

The nanoparticles can attach to the solid matrix, $C_{n*}$, in a reversible, $C_{n*}^{(r)}$ [$M_n/M_{sm}$], and/or irreversible, $C_{n*}^{(i)}$ [$M_n/M_{sm}$] manner. Therefore, the corresponding nanoparticle accumulation term in Equation (1) can be expressed as [41]:

$$\frac{\rho_b}{\theta}\frac{\partial C_{n*}(t,x)}{\partial t} = \frac{\rho_b}{\theta}\left[\frac{\partial C_{n*}^{(r)}(t,x)}{\partial t} + \frac{\partial C_{n*}^{(i)}(t,x)}{\partial t}\right] \tag{2}$$

The reversible accumulation term is given by [40,42]:

$$\frac{\rho_b}{\theta}\frac{\partial C_{n*}^{(r)}(t,x)}{\partial t} = r_{n-n*(r)}C_n(t,x) - r_{n*(r)-n}\frac{\rho_b}{\theta}C_{n*}^{(r)}(t,x) \tag{3}$$

where $r_{n*(r)-n}$ [1/t] is the rate coefficient of nanoparticle detachment from the solid matrix, and $r_{n-n*(r)}$ [1/t] is the reversible rate coefficient of nanoparticle attachment onto the solid matrix. The irreversible accumulation term is given by [43]:

$$\frac{\rho_b}{\theta}\frac{\partial C_{n*}^{(i)}(t,x)}{\partial t} = r_{n-n*(i)}C_n(t,x) \tag{4}$$

where $r_{n-n*(i)}$ is the forward rate coefficient of irreversible nanoparticle attachment onto the solid matrix.

The one-dimensional transport of solutes in water-saturated, homogeneous porous media with the one-directional uniform flow, accounting for solute sorption onto (i) suspended nanoparticles, (ii) solid matrix, and (iii) nanoparticles already attached to the solid matrix, is governed by the following partial differential equation [1,41,44,45]:

$$\frac{\partial}{\partial t}\left(C_s + \frac{\rho_b}{\theta}C_{s*} + C_nC_{ns} + \frac{\rho_b}{\theta}C_{n*}C_{n*s*}\right) = D_s\frac{\partial^2 C_s}{\partial x^2} + D_{ns}\frac{\partial^2}{\partial x^2}(C_nC_{ns})$$
$$-U_x\frac{\partial}{\partial x}\left(C_s + C_nC_{ns}\right) + F_s(t,x) \tag{5}$$

where $D_s$ [L²/t] is the longitudinal hydrodynamic dispersion coefficient of the suspended solutes; $D_{ns}$ [L²/t] is the longitudinal hydrodynamic dispersion coefficient of nanoparticle-solute complexes; and $F_s$ [$M_v/L^3t$] is a general form of the virus source configuration.

Following a similar approach to the two-site nanoparticle attachment (see Equations (3) and (4)), it is assumed here that both solutes and solute-nanoparticles complexes may reversibly and/or irreversibly sorb or attach onto the solid matrix. Therefore, the solute accumulation rate can be written as:

$$\frac{\rho_b}{\theta}\frac{\partial C_{s*}(t,x)}{\partial t} = \frac{\rho_b}{\theta}\left[\frac{\partial C_{s*}^{(r)}(t,x)}{\partial t} + \frac{\partial C_{s*}^{(i)}(t,x)}{\partial t}\right] \tag{6}$$

where the $C_{s*}^{(r)}$ [$M_n/M_{sm}$] is the reversible sorbed solute concentration onto the solid matrix and $C_{s*}^{(i)}$ [$M_n/M_{sm}$] is the irreversible sorbed solute concentration onto the solid matrix. The reversible term on Equation (6) is given by given [40,42]:

$$\frac{\rho_b}{\theta}\frac{\partial C_{s*}^{(r)}(t,x)}{\partial t} = r_{s-s*(r)}C_s(t,x) - r_{s*(r)-s}\frac{\rho_b}{\theta}C_{s*}^{(r)}(t,x) \tag{7}$$

where $r_{s-s*(r)}$ [1/t] is the reversible solute sorption rate coefficient onto the solid matrix; and $r_{s*(r)-s}$ [1/t] is the solute desorption rate coefficient from the solid matrix. The irreversible accumulation term of Equation (6) is given by [43]:

$$\frac{\rho_b}{\theta}\frac{\partial C_{s*}^{(i)}(t,x)}{\partial t} = r_{s-s*(i)}C_s(t,x) \tag{8}$$

where $r_{s-s*(i)}$ is the forward rate coefficient of irreversible solute sorption onto the solid matrix.

Furthermore, the accumulation term of solute-nanoparticle complexes onto the solid matrix present in Equation (5) is given as:

$$\frac{\rho_b}{\theta}\frac{\partial C_{n*}C_{n*s*}(t,x)}{\partial t} = \frac{\rho_b}{\theta}\left[\frac{\partial C_{n*}C_{n*s*}^{(r)}(t,x)}{\partial t} + \frac{\partial C_{n*}C_{n*s*}^{(i)}(t,x)}{\partial t}\right] \tag{9}$$

where the $C_{n*s*}^{(r)}$ [$M_s/M_n$] is the reversibly attached concentration of solute-nanoparticle complexes onto the solid matrix, and $C_{n*s*}^{(i)}$ [$M_s/M_n$] is the irreversibly attached concentration of solute-nanoparticle complexes onto the solid matrix. The reversible accumulation term present in Equation (9) is given by [1,41,46]:

$$\frac{\rho_b}{\theta}\frac{\partial}{\partial t}(C_{n*}C_{n*s*}^{(r)}) = \frac{\rho_b}{\theta}r_{s-n*s*}(C_{n*})^2 C_s - \frac{\rho_b}{\theta}r_{n*s*-s}(C_{n*}C_{n*s*}^{(r)}) + r_{ns-n*s*}(C_n C_{ns}) - \frac{\rho_b}{\theta}r_{n*s*-ns}(C_{n*}C_{n*s*}^{(r)}) \qquad (10)$$

where $r_{s-n*s*}$ [$L^3 M_{sm}/M_n{}^2 t$] is the rate coefficient of solute sorption onto nanoparticles already attached to the solid matrix; $r_{n*s*-s}$ [$1/t$] is the rate coefficient of solute desorption from solute-nanoparticles complexes attached to the solid matrix; $r_{ns-n*s*}$ [$1/t$] is the rate coefficient of solute-nanoparticle complex attachment onto the solid matrix; and $r_{n*s*-ns}$ [$1/t$] is the rate coefficient of solute-nanoparticle complex detachment from the solid matrix. Furthermore, the irreversible accumulation term of Equation (9) can be given from:

$$\frac{\rho_b}{\theta}\frac{\partial}{\partial t}(C_{n*}C_{n*s*}^{(i)}) = K_{ns}C_n C_{ns} \qquad (11)$$

where $K_{ns}$ [$1/t$] is the irreversible rate of solute-nanoparticle attachment on the solid matrix.

The third accumulation term on the left-hand side of Equation (5) is given by [1,41,46]:

$$\frac{\partial}{\partial t}(C_n C_{ns}) = r_{s-ns}(C_n)^2 C_s - r_{ns-s}(C_n C_{ns}) + \frac{\rho_b}{\theta}r_{n*s*-ns}(C_{n*}C_{n*s*}) - r_{ns-n*s*}(C_n C_{ns}) \qquad (12)$$

where $r_{s-ns}$ [$L^6/M_n{}^2 t$] is the rate coefficient of solute sorption onto suspended nanoparticles; $r_{ns-s}$ [$1/t$] is the rate coefficient of solute desorption from suspended nanoparticles. Furthermore, it is assumed that for nanoparticle facilitated transport, the formation of $C_{n*s*}$ depends only on $C_{n*}^{(r)}$, which implies that solutes do not interact with irreversibly attached nanoparticles onto the solid matrix.

Finally, combining Equations (5), (6), (8), (9), and (11) yields the following governing equation that describes the cotransport of solutes and nanoparticles:

$$\frac{\partial}{\partial t}\left(C_s + \frac{\rho_b}{\theta}C_{s*}^{(r)} + C_n C_{ns} + \frac{\rho_b}{\theta}C_{n*}C_{n*s*}^{(r)}\right) = D_s\frac{\partial^2 C_s}{\partial x^2} + D_{ns}\frac{\partial^2}{\partial x^2}(C_n C_{ns})$$
$$-U_x\frac{\partial}{\partial x}(C_s + C_n C_{ns}) - r_{s-s*(i)}C_s(t,x,y,z) - K_{ns}C_n C_{ns} + F_s(t,x,y,z) \qquad (13)$$

The general functional form of the source configuration that can be used by both nanoparticles and solutes substances can be written as [42]:

$$F_i(t,x) = G_i(t)W(x) \qquad (14)$$

where the subscript i represents either nanoparticles (i = n) or solute substance (i = s); $W(x)$ [$1/L^3$] describes the source physical geometry; and $G_i(t)$ [$M_i/t$] is the mass release function for a point source of species i. More information about different expressions of source configuration and also about the necessary initial and boundary conditions can be found in the work of Katzourakis and Chrysikopoulos [39].

### 3.2. The Fitting Process

The transport breakthrough curves were fitted with the nonlinear least squares regression software ColloidFit [47], which incorporates a model for the transport of suspended particles in water-saturated, one-dimensional, homogeneous porous media under uniform flow, accounting for nonequilibrium reversible attachment and gravity effects (Equations (1)–(4)). ColloidFit internally employs Pest [48], a stand-alone software package that uses the Gauss Marquardt Levenberg with Broyden Jacobian updating method. This numerical method is capable of estimating multiple unknown model parameters together with their 95% confidence intervals, even for nonlinear models. Furthermore, for the cotransport of TM-TiO₂ experiments, the mathematical model developed here

(Equations (1)–(4), (7), and (10)–(13)) was solved numerically, and in conjunction with Pest, the required fittings were performed.

The fitting approach was as follows: initially, the TM transport experiments were fitted with ColloidFit. The parameters $D_s$, $r_{s-s*^{(r)}}$, $r_{s*^{(r)}-s}$, and $r_{s-s*^{(i)}}$ were calculated and listed in Table 2. Subsequently, the TiO$_2$ breakthrough curves collected from the TM-TiO$_2$ cotransport experiments were also fitted with ColloidFit, and the parameters determined are: $D_n$, $r_{n-n*^{(r)}}$, $r_{n*^{(r)}-n}$, and $r_{n-n*^{(i)}}$. This simplified approach is based on the assumption that, due to their size, the TiO$_2$ nanoparticles are practically not affected by the presence of the TM solutes. This approach is similar to that proposed by Katzourakis and Chrysikopoulos [41]. Finally, the TM breakthrough curves from the TM-TiO$_2$ cotransport experiments were fitted with Equations (1)–(4), (7), and (10)–(13).

**Table 2.** Transport and cotransport measured and fitted parameter values for Q = 1 mL/min.

| Experimental Conditions | pH | | | | Ionic Strength (mM) | | | |
|---|---|---|---|---|---|---|---|---|
| | 3 | 5 | 7 | 10 | 1 | 10 | 50 | 100 |
| **Transport parameter values for TM** | | | | | | | | |
| $M_r$ (%) | 99.0 | 100 | 100 | 100 | 97.2 | 95.3 | 99.5 | 99.3 |
| U (cm/min) | 0.53 | 0.54 | 0.54 | 0.54 | 0.53 | 0.54 | 0.55 | 0.54 |
| $t_p$ (min) | 240 | 242 | 242 | 234 | 231 | 237 | 239 | 234 |
| θ (-) | 0.38 | 0.38 | 0.39 | 0.38 | 0.38 | 0.37 | 0.37 | 0.38 |
| $D_s$ (cm/min) | 0.26 | 0.24 | 0.29 | 0.30 | 0.22 | 0.23 | 0.27 | 0.29 |
| $r_{s-s*^{(r)}}$ (1/min) | 0.0055 | 0.0032 | 0.0022 | 0.0019 | 0.0036 | 0.0038 | 0.0058 | 0.0060 |
| $r_{s*^{(r)}-s}$ (1/min) | 0.0204 | 0.0182 | 0.0136 | 0.0060 | 0.0117 | 0.0136 | 0.0165 | 0.0171 |
| $r_{s-s*^{(i)}}$ (1/min) | 0.00086 | 0 | 0 | 0 | 0.00037 | 0.00072 | 0.00090 | 0.00094 |
| **Cotransport parameter values for TM** | | | | | | | | |
| $M_r$ (%) | 92.9 | 95.2 | 98.6 | 90.9 | 57.9 | 54.3 | 34.3 | 31.5 |
| U (cm/min) | 0.54 | 0.53 | 0.54 | 0.54 | 0.54 | 0.55 | 0.54 | 0.53 |
| $t_p$ (min) | 231 | 229 | 232 | 237 | 235 | 228 | 222 | 224 |
| θ (-) | 0.37 | 0.38 | 0.38 | 0.38 | 0.38 | 0.37 | 0.38 | 0.38 |
| $D_s$ (cm/min) | 0.26 | 0.24 | 0.29 | 0.30 | 0.22 | 0.23 | 0.27 | 0.29 |
| $r_{s-ns}$ ($L^6/M_n^2 t$) | 0.00064 | 0.0020 | 0.0023 | 0.0071 | 0.9000 | 0.2870 | - | - |
| $r_{ns-s}$ (1/min) | 0.0102 | 0.0296 | 0.0585 | 0.0654 | 0.7300 | 0.1780 | - | - |
| $r_{n*s*-s}$ (1/min) | 0.0085 | 0.0004 | 0.0786 | 0.0834 | 0.2820 | 0.0003 | - | - |
| **Cotransport parameter values for TiO$_2$ nanoparticles** | | | | | | | | |
| $M_r$ (%) | 92.8 | 94.6 | 100 | 100 | 61.5 | 53.3 | 3.29 | 2.81 |
| U (cm/min) | 0.54 | 0.53 | 0.54 | 0.54 | 0.54 | 0.55 | 0.54 | 0.53 |
| $t_p$ (min) | 231 | 229 | 232 | 237 | 235 | 228 | 222 | 224 |
| θ (-) | 0.37 | 0.38 | 0.38 | 0.38 | 0.38 | 0.37 | 0.38 | 0.38 |
| $D_n$ (cm/min) | 0.26 | 0.23 | 0.29 | 0.30 | 0.22 | 0.5 | - | - |
| $r_{n-n*^{(r)}}$ (1/min) | 0.001 | 0.0008 | 0.0004 | 0.0007 | 0.0015 | 0.0044 | - | - |
| $r_{n*^{(r)}-n}$ (1/min) | 0.0185 | 0.0205 | 0.0111 | 0.0053 | 0.0127 | 0.0257 | - | - |
| $r_{n-n*^{(i)}}$ (1/min) | 0.0014 | 0.0009 | - | 0.0009 | 0.0089 | 0.0110 | - | - |

Due to the large number of parameters involved in the current model, and in order to avoid over-fitting, which could produce non-unique results, several assumptions were made which allowed the use of existing parameters obtained from previous studies. The

parameters values for $D_s$, $r_{s-s*(r)}$, $r_{s*(r)-s}$, and $r_{s-s*(i)}$ required by the cotransport model, were obtained from the corresponding TM transport experiments, which were carried out under the same experimental conditions (Is and pH). Given that nanoparticles and nanoparticle-solute complexes have similar sizes, it was assumed that the parameters $r_{ns-n*s*}$, $r_{n*s*-ns}$, and $K_{ns}$ were identical to the $r_{n-n*}$, $r_{n*-n}$, and $r_{n-n*(i)}$, respectively. Similarly, the sorption rate of solutes onto previously attached nano-solute complexes ($r_{s-n*s*}$), is expected to be similar to the sorption rate of solutes onto the solid matrix, $r_{s-s*}$. Consequently, solutes sorb with the same rate onto both the solid matrix and complexes previously attached to the solid matrix. Based on the above assumptions, only three parameters of the cotransport model ($r_{s-ns}$, $r_{s-ns}$, and $r_{s-ns}$) should be fitted. These fitted parameters are listed in Table 2.

*3.3. Additional Theoretical Calculations*

The recovered mass ($M_r$ [-]) for both TM and $TiO_2$ NPs at the column outlet was determined with the application of the following mathematical relationship [35]:

$$M_r(L) = \frac{m_0(L)}{C_0 t_p} \tag{15}$$

where $C_0$ [$M/L^3$] is the aqueous phase concentration of either TM or $TiO_2$ NPs; $t_p$ [t] is the broad pulse duration; $m_0$ [$tM/L^3$] is the zeroth absolute temporal moment that quantifies the total mass in the concentration distribution curve [46]:

$$m_0(L) = \int_0^\infty C(L, t)dt \tag{16}$$

where C [$M/L^3$] is the aqueous phase concentration of either TM or $TiO_2$ NPs, L [L] is the length of the packed column; and t [t] is time. All $M_r$ estimates were obtained using Equations (14) and (15), as determined by the software ColloidFit [47].

**4. Results and Discussion**

The batch kinetic experiments of TM sorption onto quartz sand, under dynamic and static conditions at four different pH values (pH = 3, 5, 7, and 10) and for four different ionic strength values (Is = 1, 10, 50, and 100 mM), with a TM initial concentration of $C_0$ = 10 mg/L, at room temperature, are presented in Figure 2. Clearly, the experimental data suggested that during static conditions, neither the pH nor Is significantly affected the sorption of TM onto quartz sand. As expected, under dynamic conditions, the amount of TM sorbed onto quartz sand increased slightly with time due to agitation, but the sorption of TM onto quartz sand was insensitive to both pH and Is variations. These findings are in agreement with the work by Flores et al. [49], who reported that the sorption of TM onto montmorillonite is insignificant (<5%), and the work by Wauchope et al. [50], who reported that TM does not sorb strongly onto soil particles.

The normalized TM breakthrough concentrations ($C/C_0$) and the corresponding fitted curves as a function of time are presented in Figure 3 for the transport experiments in water-saturated columns packed with quartz sand. The effect of four different ionic strength values (Is = 1, 10, 50, and 100 mM) is shown in Figure 3a–d, whereas the effect of four different pH values (pH = 3, 5, 7, and 10) is shown in Figure 3e–h. All breakthrough curves were successfully fitted with the nonlinear least squares regression software ColloidFit. The experimental conditions, together with the estimated mass recoveries and the fitted parameters, for each case considered in this study, are listed in Table 2.

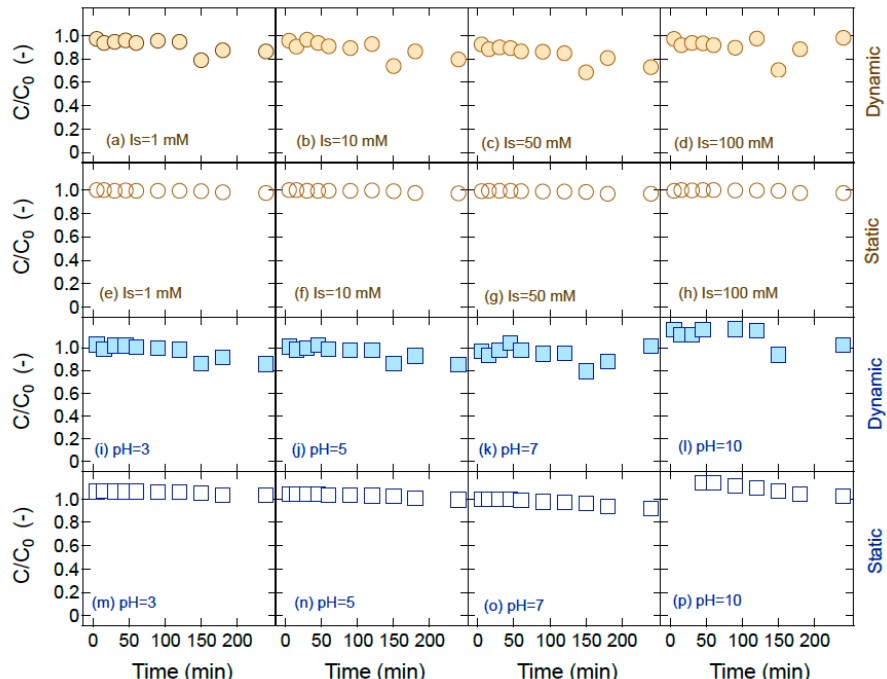

**Figure 2.** TM normalized concentrations during the batch adsorption experiments onto quartz sand under: dynamic conditions and four different Is values (**a**–**d**), static conditions and four different Is values (**e**–**h**), dynamic conditions and four different pH values (**i**–**l**), and static conditions and four different Is values (**m**–**p**). Here the initial TM concentration is $C_0$ = 10 mg/L.

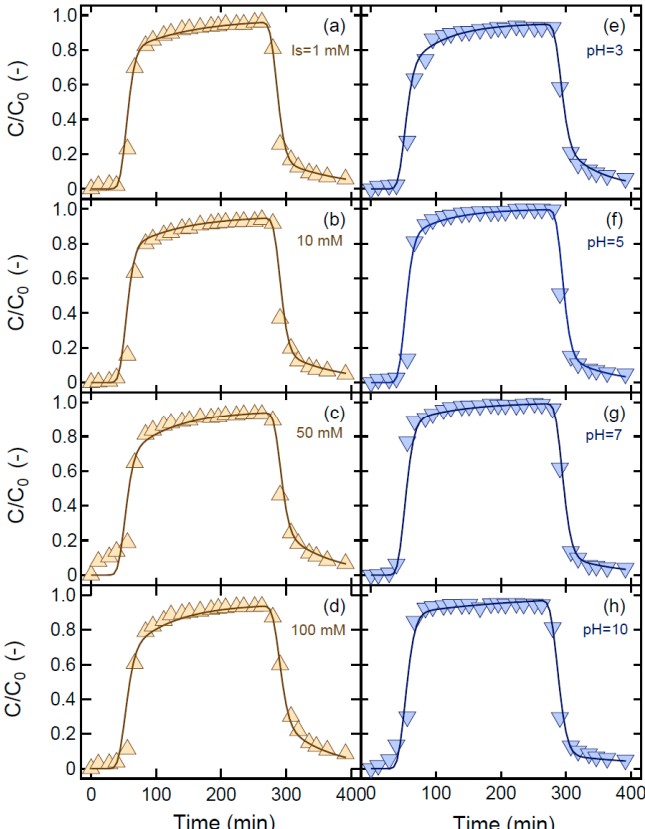

**Figure 3.** TM normalized breakthrough concentrations (symbols) at four different Is values (**a**–**d**) and four different pH values (**e**–**h**) collected at the exit of the water-saturated columns packed with quartz sand, together with the best fitted model simulations (solid curves).

Mass recoveries and maximum concentrations for TM with changing Is remained practically constant. However, the lowest mass recoveries were obtained for Is = 1 mM (97.2%) and for Is = 10 mM (95.3%) (see Table 2). Similarly, the mass recoveries for TM with changing pH did not exhibit substantial variability. Additionally, mass recoveries approached 100% for some pH values.

The fitted parameter values for the TM transport experimental data suggested that increasing the pH caused the reversible sorption, $r_{s-s*(r)}$, and desorption, $r_{s*(r)-s}$, rates to decrease (see Table 2). Furthermore, increasing Is caused both reversible and irreversible sorption rates ($r_{s-s*(r)}$, $r_{s-s*(i)}$) to increase. Therefore, for the particular solution chemistry, increasing the pH hinders the sorption rate, while increasing the Is enhances it. These results are in agreement with previous sorption studies [51–53].

The collected breakthrough experimental data, together with the model fitted curves for the TM and TiO$_2$ cotransport experiments, for several different ionic strengths (Is = 1, 10, 50, and 100 mM) and pH values (pH = 3, 5, 7, and 10) as a function of time, are presented in Figures 4 and 5, respectively. Note that the mass recovery values for TM were considerably lower in the presence of TiO$_2$ compared to the single transport experiments, where $M_r$ = 100% was observed. The measured zeta potentials for TiO$_2$ in the various experiments, which are listed in Table 1, suggested that the zeta potential for TiO$_2$ was affected during the various experiments. The observed negative increase in the zeta potential, and thus an increase in stability, was not only due to the increase in pH but also due to possible TM sorption onto TiO$_2$. The observed reduced mass recovery of TM during the cotransport with TiO$_2$ could also be explained by the possible sorption of TM onto TiO$_2$ nanoparticles. Previous studies have shown that fungicides can act as capping agents for metal nanoparticles. Malandrakis et al. [54] demonstrated such a capping effect of ZnO nanoparticles when applied in combination with the fungicide boscalid against the plant pathogen *Alternaria alternata*. A similar interaction between TM and silver nanoparticles was reported in a study by Zheng et al. [55], who developed a colorimetric array for the detection of TM adsorbed onto silver nanoparticles. The fitted parameter values for TiO$_2$ and TM concentrations are listed in Table 2. The mathematical model presented in this work successfully fitted the experimental breakthrough data. It should be noted, however, that the experiments for Is = 50 and 100 mM (see Figure 5g,h) caused extreme retention of TiO$_2$ inside the column ($M_r$ < 5%). Consequently, it was very hard for the cotransport model to produce meaningful and unique parameter values. Therefore, these two experimental data sets were excluded from the fitting process.

Both models (simple transport and cotransport) considered in this study provided excellent fits to the experimental data; however, none of them accounts for aggregation. Due to the nature of TiO$_2$ nanoparticles, increasing the ionic strength causes them to aggregate and increase their size. From Table 1, it is evident that for values of ionic strength in the range of 1 to 50 mM, the average hydrodynamic diameter size ($d_H$) fluctuates around 351 nm, but for Is = 100 mM, the $d_H$ rapidly increases to 1004 nm. This is a consequence of particle aggregation that changes the physical characteristics of particle transport and dramatically reduces the mass recovery ratio $M_r$. Aggregating nanoparticles require specialized models [56] for their simulation. This is the reason that the experimental data for high ionic strength values (Is = 50 & 100 mM) were excluded from the fitting process.

The estimated mass recovered for TM at Is = 100 mM was $M_r$ = 31.5% and for Is = 50 mM was $M_r$ = 34.3%, suggesting that TM removal from soil could be enhanced in the presence of TiO$_2$ nanoparticles. These observations are in agreement with the fitting results of this study, which indicated that increasing ionic strength contributes to the increase of both reversible, $r_{n-n*(r)}$, and irreversible, $r_{n-n*(i)}$, attachment rates. The effect of salt concentration on pesticide sorption is complex. Based on the diffuse double-layer theory, ions that form outer-sphere surface complexes show decreasing adsorption with increasing ionic strength, while ions that form inner-sphere surface complexes show little ionic strength dependence or show increasing adsorption with increasing ionic strength [57,58]. In previous studies, it has been observed that negatively charged TiO$_2$ nanoparticles are attached to

positively charged sand [59]. Additionally, according to Chrysikopoulos and Fountouli [60], the presence of NaCl affected substantially the transport of $TiO_2$ nanoparticles, yielding a reduction in $M_r$, which is also consistent with the results of the present study.

The results of this study suggest that by increasing the solution pH the mass recovery for both TM and $TiO_2$ was enhanced (see Table 2), and the reversible attachment rate ($r_{n-n*(r)}$) for $TiO_2$ was decreased. Both of these observations are valid for all pH values examined in this study except for the experiment at pH = 10, where the results were exactly the opposite (TM mass recovery decreased, and $r_{n-n*(r)}$ increased). This unexpected result might be due to the strong sorption rate of TM onto $TiO_2$ ($r_{s-ns}$) observed at pH = 10 (see Table 2). It should be noted that, with increasing the pH, the sorption rate of TM onto the solid matrix ($r_{s-s*(r)}$) was decreased (see Table 2). However, in the presence of $TiO_2$ nanoparticles, at high pH values the sorption rate of TM onto $TiO_2$ particles was increased. The sorption rate $r_{s-ns}$ followed an increasing trend with increasing pH (see Table 2) and suggested that $TiO_2$ nanoparticles facilitated TM transport progressively more with increasing pH.

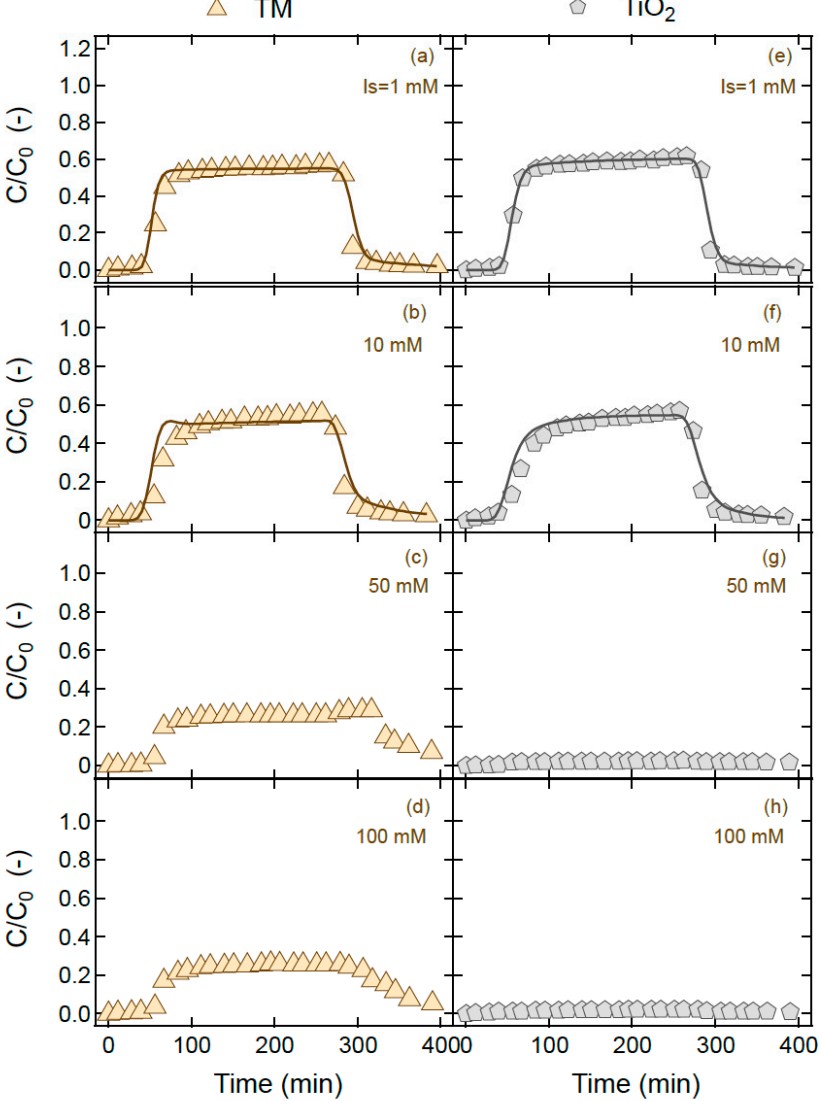

**Figure 4.** Breakthrough data (symbols) for the cotransport of TM (triangles) (**a**–**d**) and $TiO_2$ (pentagons) (**e**–**h**), together with the corresponding fitted curves (solid curves), in a column packed with quartz sand, at 4 different ionic strengths (Is = 1, 10, 50, and 100 mM) as a function of time.

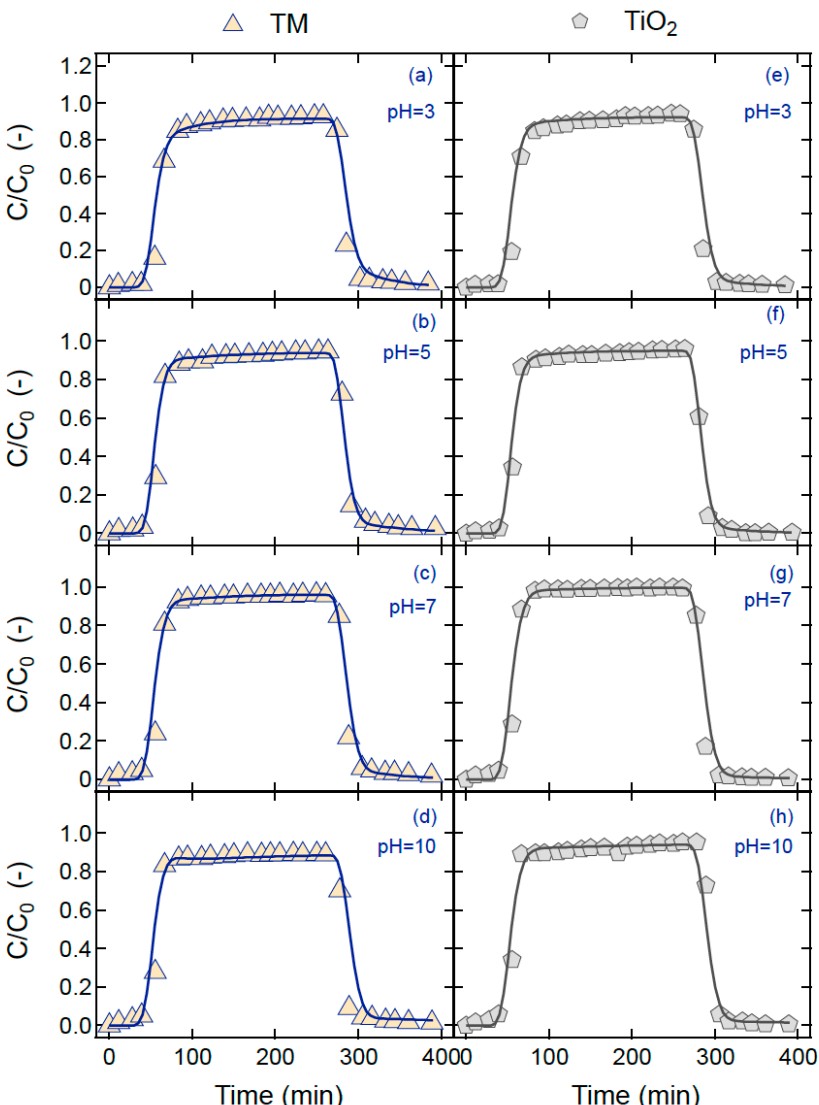

**Figure 5.** Breakthrough data (symbols) for the cotransport of TM (triangles) (**a**–**d**) and TiO$_2$ (pentagons) (**e**–**h**), together with the corresponding fitted curves (solid curves), in a column packed with quartz sand, at 4 different pH values (pH = 3, 5, 7, and 10) as a function of time.

Nanoparticles can be produced in various sizes and shapes. The initial particle diameter is a very important factor that affects nanoparticle migration. Further increase in particle size due to aggregation may intensify particle attachment or reduce it [56]. The Filtration Theory is often used to quantify the effect of size increase on particle attachment and to predict whether aggregation may ultimately enhance particle transport or hinder it. Aggregation may cause nanoparticles to undergo straining, which effectively restricts them from entering the smaller pores of the solid matrix and, in turn, may increase their retention. Moreover, particle size increase may also affect the effective dispersivity of aggregating nanoparticles. Larger particles may exhibit early breakthroughs and increased dispersivity due to possible reduction in effective porosity and exclusion from lower interstitial velocity regions [61]. Finally, due to the sorption of TM onto TiO$_2$ nanoparticles, all of the above factors, which are expected to enhance nanoparticle transport, are expected to also enhance the TM transport, while the inverse is also true.

## 5. Conclusions

The results from the present experiments focused on the transport of TM and cotransport of TiO$_2$ and TM in a water-saturated column packed with quartz sand under various

ionic strength and pH conditions, suggested that increasing the solution pH: (i) reduced the sorption rate of TM onto the solid matrix, (ii) reduced the attachment rate of $TiO_2$ nanoparticles onto the solid matrix, (iii) increased the sorption rate of TM onto $TiO_2$ nanoparticles, and (iv) increased the mass recovery of both TM and $TiO_2$. On the contrary, increasing the ionic strength yielded: (i) increased sorption of TM onto the solid matrix, (ii) increased attachment rate of $TiO_2$ particles onto the solid matrix, and (iii) reduced mass recovery of both $TiO_2$ particles and TM solutes. Furthermore, for the cotransport case, under the experimental conditions of pH = 5.1 and Is = 100 mM, it was shown that the mass retention of TM by the packed column was highest or equivalent TM mass recovery was lowest ($M_r$ = 31.5%).

The presence of $TiO_2$ nanoparticles for Is = 100 mM yielded a 67.8% reduction in TM mass recovery, suggesting that $TiO_2$ nanoparticles can be used to enhance the removal of TM from the soil. Similarly, when considering the transport of $TiO_2$ nanoparticles, it was shown that increasing the ionic strength from 1 to 100 mM dramatically decreased their mass recovery, highlighting their sensitivity to ionic strength. Finally, it is evident from the current study that the solution pH and ionic strength can affect the TM transport characteristics, with the latter one having more profound effects in the presence of $TiO_2$ nanoparticles.

**Author Contributions:** A.S.S. worked on the experimental design, collected, and analyzed the data, and wrote the first draft of the paper. V.E.K. Focused on the modeling aspects of this work and performed all nonlinear least squares regressions. F.F. contributed to the presentation and analysis of results, manuscript reviewing and editing. A.A.M. contributed to the methodology, experimental design, and editing of the manuscript. C.V.C. conceptualized and supervised the research and contributed to the writing-reviewing and editing of the manuscript. All authors have read and agreed to the published version of the manuscript.

**Funding:** This project received funding from the Partnership for Research and Innovation in the Mediterranean Area (PRIMA), under grant agreement number: 1923-InTheMED, as well as from Khalifa University Grant/Award Number: FSU-2023-012.

**Data Availability Statement:** The data presented in this study are available on request from the corresponding author.

**Acknowledgments:** The authors are thankful to Rika Sarika for valuable laboratory assistance.

**Conflicts of Interest:** The authors declare that there is no conflict of interest.

## Nomenclature

| | |
|---|---|
| C | aqueous phase concentration, $M/L^3$ |
| $C_n$ | aqueous phase concentration of suspended nanoparticles, $M_n/L^3$. |
| $C_{n*}$ | concentration of nanoparticles attached onto the solid matrix, $M_n/M_{sm}$. |
| $C_s$ | aqueous phase solute concentration, $M_s/L^3$. |
| $C_{s*}$ | concentration of solutes sorbed onto the solid matrix, $M_s/M_{sm}$. |
| $C_{ns}$ | concentration of suspended solute-nanoparticle complex, $M_s/M_n$. |
| $C_{n*s*}$ | concentration of solute-nanoparticle complex attached onto the solid matrix, $M_s/M_n$. |
| $C_{n*}^{(i)}$ | concentration of nanoparticles irreversibly attached onto the solid matrix, $M_n/M_{sm}$. |
| $C_{n*}^{(r)}$ | concentration of nanoparticles reversibly attached onto the solid matrix, $M_n/M_{sm}$. |
| $C_0$ | initial aqueous phase solute concentration, $M_s/L$ |
| $d_H$ | hydrodynamic diameter, L |
| $D_i$ | hydrodynamic dispersion coefficient of species i, $L^2/t$. |
| $F_i$ | general form of the source configuration of species i, $M_i/L^3t$. |
| $G_i(t)$ | mass release function of species i (point source), $M_i/t$. |
| Is | ionic strength, mM |
| $K_{ns}$ | rate of irreversible solute-nanoparticle complex attachment onto the solid matrix, $1/t$ |
| L | length, L. |
| $m_0$ | zeroth absolute temporal moment, $tM/L^3$ |

| | |
|---|---|
| $M_n$ | mass of nanoparticles, $M_n$. |
| $M_{sm}$ | mass of the solid matrix, $M_{sm}$. |
| $M_s$ | mass of solutes, $M_s$. |
| $M_r$ | ratio of recovered mass, [-] |
| $r_{n-n*^{(i)}}$ | rate of irreversible nanoparticle attachment onto the solid matrix, $1/t$. |
| $r_{n-n*^{(r)}}$ | rate of reversible nanoparticle attachment onto the solid matrix, $1/t$. |
| $r_{n*^{(r)}-n}$ | rate of reversible nanoparticle detachment from the solid matrix, $1/t$. |
| $r_{s-s*^{(i)}}$ | rate of irreversible solute sorption onto the solid matrix, $1/t$. |
| $r_{s-s*^{(r)}}$ | rate of reversible solute sorption onto the solid matrix, $1/t$. |
| $r_{s*^{(r)}-s}$ | rate of reversible solute desorption from the solid matrix, $1/t$. |
| $r_{s-ns}$ | rate of solute sorption onto suspended nanoparticles, $L^6/M_n^2 t$. |
| $r_{ns-s}$ | rate of solute desorption from suspended nanoparticles, $1/t$. |
| $r_{s-n*s*}$ | rate of solute sorption onto nanoparticles already attached onto the solid matrix, $L^3 M_{sm}/M_n^2 t$. |
| $r_{ns-n*s*}$ | rate of solute-nanoparticle complex attachment onto the solid matrix, $1/t$. |
| $r_{ns-s}$ | rate of solute desorption from suspended nanoparticles, $1/t$. |
| $r_{n*s*-ns}$ | rate of solute-nanoparticle complex detachment from the solid matrix, $1/t$. |
| $t$ | time, $t$. |
| $t_p$ | source duration time period, $t$. |
| $U$ | interstitial velocity, $L/t$. |
| $W$ | characterizes the source physical geometry (point source), $1/L^3$. |
| $x$ | Cartesian coordinate, $L$. |
| **Greek Letters** | |
| $\Theta$ | porosity, $(L^3 \text{ voids})/(L^3 \text{ solid matrix})$. |
| $\zeta$ | zeta potential, $V$ |
| $\rho$ | bulk density of the solid matrix, $M_{sm}/L^3$ |

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
