# Peer review of "Transport of Thiophanate Methyl in Porous Media in the Presence of Titanium Dioxide Nanoparticles"

_water, doi:10.3390/w15071415_

Round 1
Reviewer 1 Report
Dear Authors, please elaborate your Introduction and Conclusions. Introductions requires more insight on the novelty of research and reacher citations. There should be clearance on what was done in this field of research, with more detailed info and what is not researched, barriers, etc. Conclusions are way too general.
Author Response
We thank Reviewer 1 for the time and effort to review our work, and for the positive comments and constructive criticism. Based on the Reviewer’s recommendations, the following changes were made in the revised manuscript (highlighted in yellow in the revised manuscript):
The introduction was expanded to highlight the novelty of the current study. It is now stated that despite various approaches of pesticide removal using nanoparticles presented in the literature, no data is available regarding the efficiency of TiO2 nanoparticles in TM removal, under different pH and Is conditions.
Furthermore, the collected TM and TiO2 experimental breakthrough curves were fit with a numerical model previously published in the literature, which was properly modified to account for the irreversible sorption of TM onto TiO2 nanoparticles. It should be noted that numerus additional references were added and discussed in the revised version of the introduction section.
The conclusion section was also extended in the revised manuscript. It now contains specific information regarding the effect of the TiO2 presence on the TM mass recovery ratio, which can be reduced considerably for the same experimental conditions. Furthermore, the sensitivity of TiO2 nanoparticles to the ionic strength was also discussed. As a final remark, it is now clearly recognized that the pH and the ionic strength parameters can affect the transport of TM. Specifically, it was stated in the revised manuscript that the presence of TiO2 nanoparticles, for Is=100 mM, yielded a 67.8% reduction in TM mass recovery, suggesting that TiO2 nanoparticles can be used to enhance the removal of TM from soil. Similarly, when considering the transport of TiO2 nanoparticles, it was shown that increasing the ionic strength from 1 to 100 mM dramatically decreased their mass recovery, highlighting their sensitivity to ionic strength. Finally, it is evident from the current study that the solution pH and ionic strength can affect the TM transport characteristics with the latter one having more profound effects in the presence of TiO2nanoparticles.
We have considered carefully all comments and suggestions provided by Reviewer 1. Certainly, the manuscript has improved by the changes made in response to the suggestions provided. We are thankful, and we trust that our revised manuscript is now acceptable for publication in Water (Switzerland).
Reviewer 2 Report
Dear Editor: This paper is worth to be published here. But the authors shall give the chemical structure of thiophanate methyl and propose the interaction model as well as the finding presented in the Abstract section.
Author Response
We thank Reviewer 2 for the time and effort to review our work, and for the positive comments and constructive criticism. Based on the Reviewer’s recommendations, the following changes were made in the revised manuscript (highlighted in yellow in the revised manuscript):
A new figure with the chemical structure of thiophanate methyl (see Figure 1) was added in the revised manuscript, as suggested by the Reviewer.
We kindly note that it was clearly stated in the manuscript that based on the diffuse double-layer theory, ions that form outer-sphere surface complexes show decreasing adsorption with increasing ionic strength, while ions that form inner-sphere surface complexes show little ionic strength dependence or show increasing adsorption with increasing ionic strength. In previous studies it has been observed that negatively charged TiO2 nanoparticles attached onto positively charged sand. Also, it was reported that the presence of NaCl affected substantially the transport of TiO2nanoparticles, yielding a reduction in Mr, which is also consistent with the results of the present study.
Furthermore, it was clearly stated in the manuscript that the results of this study suggest that by increasing the solution pH the mass recovery for both TM and TiO2 was enhanced (see Table 2), and the reversible attachment rate for TiO2 was decreased. Also, the conclusion section was expanded to explain that Nanoparticles can be produced in various sizes and shapes. The initial particle diameter is a very important factor that affects nanoparticle migration. Further increase in particle size due to aggregation, may intensify particle attachment or reduce it. The Filtration Theory is often used to quantify the effect of size increase on particle attachment, and to predict whether aggregation may ultimately enhance particle transport or hinder it. Aggregation may cause nanoparticles to undergo straining, which effectively restricts them from entering the smaller pores of the solid matrix and in turn may increase their retention. Moreover, particle size increase may also affect the effective dispersivity of aggregating nanoparticles. Larger particles may exhibit early breakthrough and increased dispersivity due to possible reduction in effective porosity and exclusion from lower interstitial velocity regions. Finally, due to the sorption of TM onto TiO2nanoparticles all of the above factors, which are expected to enhance nanoparticle transport, are expected to also enhance the TM transport, while the inverse is also true.
We have considered carefully all comments and suggestions provided by Reviewer 2. Certainly, the manuscript has improved by the changes made in response to the suggestions provided. We are thankful, and we trust that our revised manuscript is now acceptable for publication in Water (Switzerland).
Reviewer 3 Report
The paper "Transport of thiophanate methyl in porous media in the presence of titanium dioxide nanoparticles” is interesting.
Describe the methods and materials that need to be referenced in the statistical data processing in greater detail. Describing experiments, systems, and results is necessary before analyzing them. Extending the conclusions, what are the additional topics required for review and Implementation options?
Author Response
We thank Reviewer 3 for the time and effort to review our work, and for the positive comments and constructive criticism. Based on the Reviewer’s recommendations, the following changes were made in the revised manuscript (highlighted in yellow in the revised manuscript):
The section describing the fitting process was expanded considerably. In particular, it was clarified that both the numerical models used (simple transport and cotransport), employed the Pest software package. Pest is a standalone software capable of estimating multiple unknown parameters together along with their 95% confidence. Pest uses the Gauss Marquardt Levenberg with Broyden Jacobian updating method.
The conclusion section was also extended to include specific information regarding the effect of the TiO2presence on the TM mass recovery ratio. Furthermore, remarks were added to clearly recognize that the pH and the ionic strength parameters can affect the transport of TM. The revised conclusion section contains specific information regarding the effect of the TiO2 presence on the TM mass recovery ratio, which can be reduced considerably for the same experimental conditions. Furthermore, the sensitivity of TiO2 nanoparticles to the ionic strength was also discussed. As a final remark, it is now clearly recognized that the pH and the ionic strength parameters can affect the transport of TM. Specifically, it was stated in the revised manuscript that the presence of TiO2 nanoparticles, for Is=100 mM, yielded a 67.8% reduction in TM mass recovery, suggesting that TiO2 nanoparticles can be used to enhance the removal of TM from soil. Similarly, when considering the transport of TiO2 nanoparticles, it was shown that increasing the ionic strength from 1 to 100 mM dramatically decreased their mass recovery, highlighting their sensitivity to ionic strength. Finally, it is evident from the current study that the solution pH and ionic strength can affect the TM transport characteristics with the latter one having more profound effects in the presence of TiO2nanoparticles.
The “Results and discussion section” was expanded to include discussion on the topic of aggregation. It was clearly stated in the revised manuscript that both models (simple transport and cotransport) considered in this study provided excellent fits to the experimental data; however, none of them accounts for aggregation. Due to the nature of TiO2 nanoparticles, increasing the ionic strength causes them to aggregate and increase their size. From Table 1 it is evident that for values of ionic strength in the range 1 to 50 mM the average hydrodynamic diameter size (dH) fluctuates around 351 nm, but for Is=100 mM the dH rapidly increases to 1004 nm. This is a consequence of particle aggregation that changes the physical characteristics of particle transport and dramatically reduces the mass recovery ratio Mr. Aggregating nanoparticles require specialized models for their simulation. This is the reason that the experimental data for high ionic strength values (Is=50 & 100 mM) were excluded from the fitting process.
We have considered carefully all comments and suggestions provided by Reviewer 3. Certainly, the manuscript has improved by the changes made in response to the suggestions provided. We are thankful, and we trust that our revised manuscript is now acceptable for publication in Water (Switzerland).
Reviewer 4 Report
I have some minor suggestions.
1. What's the shape of the TiO2 NPs in the current study? The current study only used a single type of TiO2 nanoparticle, while TiO2 nanoparticles can be synthesized with different sizes, shapes and crystal structures. I suggest the authors to also include some discussion about how those parameters would affect the transport of TM.
2. I suggest the authors to add a figure or scheme to show the TM transport mechanism under various ionic strength and pH conditions for readers to better understand the information presented.
Author Response
We thank Reviewer 4 for the time and effort to review our work, and for the positive comments and constructive criticism. Based on the Reviewer’s recommendations, the following changes were made in the revised manuscript (highlighted in yellow in the revised manuscript):
Although there are several types of TiO2 nanoparticles available, in this study we used only anatase (Aldrich 637254-50G, purity N 99.9%, size < 25 nm). We kindly point out that the aim of our work was to investigate the transport of pesticide thiophanate methyl (TM) as well as the cotransport of TM and TiO2 nanoparticles in a water saturated column packed with quartz sand under various water conditions.
The “Results and discussion section” of the revised manuscript was expanded to include discussion regarding the effects of particle size on the transport of TiO2 and TM. It was clearly stated in the text that the initial particle diameter is a very important factor that affects nanoparticle migration. Further increase in particle size due to aggregation, may intensify particle attachment or reduce it. The Filtration Theory is often used to quantify the effect of size increase on particle attachment, and to predict whether aggregation may ultimately enhance particle transport or hinder it. Aggregation may cause nanoparticles to undergo straining, which effectively restricts them from entering the smaller pores of the solid matrix and in turn may increase their retention. Moreover, particle size increase may also affect the effective dispersivity of aggregating nanoparticles. Larger particles may exhibit early breakthrough and increased dispersivity due to possible reduction in effective porosity and exclusion from lower interstitial velocity regions. Finally, due to the sorption of TM onto TiO2 nanoparticles all of the above factors, which are expected to enhance nanoparticle transport, are expected to also enhance the TM transport, while the inverse is also true.
A figure that illustrates the mass recovery of TM, TM in the presence of TiO2 and TiO2 alone for low and high ionic strengths. This illustration represents the important finding of the present work and we have decided to use it as a Graphical abstract.
We have considered carefully all comments and suggestions provided by Reviewer 4. Certainly, the manuscript has improved by the changes made in response to the suggestions provided. We are thankful, and we trust that our revised manuscript is now acceptable for publication in Water (Switzerland).
Round 2
Reviewer 1 Report
Manuscript has to be submitted in the template of journal with all required formatting.